# LncRNA-MALAT1: A Key Participant in the Occurrence and Development of Cancer

**DOI:** 10.3390/molecules28052126

**Published:** 2023-02-24

**Authors:** Longhui Hao, Wenzheng Wu, Yankun Xu, Yufan Chen, Chengzhen Meng, Jingyi Yun, Xiaoyu Wang

**Affiliations:** 1College of Pharmacy, Shandong University of Traditional Chinese Medicine, Jinan 250355, China; 2College of Traditional Chinese Medicine, Shandong University of Traditional Chinese Medicine, Jinan 250355, China

**Keywords:** long non-coding RNA, lncRNA-MALAT1, cancer, mechanism, action pathway

## Abstract

LncRNAs are a group of non-coding RNA transcripts with lengths of over 200 nucleotides and can interact with DNA, RNA, and proteins to regulate gene expression of malignant tumors in human tissues. LncRNAs participate in vital processes, such as chromosomal nuclear transport in the cancerous site of human tissue, activation, and the regulation of proto-oncogenes, the differentiation of immune cells, and the regulation of the cellular immune system. The lncRNA metastasis-associated lung cancer transcript 1 (MALAT1) is reportedly involved in the occurrence and development of many cancers and serves as a biomarker and therapeutic target. These findings highlight its promising role in cancer treatment. In this article, we comprehensively summarized the structure and functions of lncRNA, notably the discoveries of lncRNA-MALAT1 in different cancers, the action mechanisms, and the ongoing research on new drug development. We believe our review would serve as a basis for further research on the pathological mechanism of lncRNA-MALAT1 in cancer and provide evidence and novel insights into its application in clinical diagnoses and treatments.

## 1. Structure of LncRNA

Approximately 2% of the genes in the entire genome sequence of organisms can code for proteins. Conversely, non-coding RNA molecules participate in various biological processes and have almost no protein translation functions [1]. The typical non-coding RNAs with regulatory functions include small interfering RNAs (siRNAs), microRNAs (miRNAs), PIWI-interacting RNAs (piRNAs), and lncRNAs. Although lncRNAs lack most of their protein-coding ability, they can be transcribed by at least 80% of mammalian genomes [2], regulating the transcription and expression of other genes in tumor cells.

Long non-coding RNAs (lncRNAs) belong to a group of non-coding RNA transcripts longer than 200 nucleotides. LncRNAs, as functional RNA molecules, participate in the gene expression of various diseases and influence biological processes and homeostasis. MALAT1, an important member of the lncRNA family, was the first to be identified as LncRNA-MALAT1. Metastasis-associated lung cancer transcript 1 (MALAT1) is closely associated with the occurrence and development of various human cancers [3]. In 2003, it was screened for several differentially expressed genes in tumor cells from patients with initial-stage non-small cell lung cancer. LncRNA-MALAT1 is an intergenic transcript that is located on human chromosome 11q13.1 and is approximately 8.7 kb long [4]. In particular, the role of lncRNA-MALAT1 in respiratory cancer, gynecological cancer, and refractory diseases has attracted extensive attention. The varied mechanisms and characteristics of lncRNA-MALAT1 in different cancers and diseases are also crucial in diagnosing related diseases, defining therapeutic targets, and developing new drugs. 

Therefore, in this article, we have comprehensively reviewed the structure, function, and role of lncRNA. Furthermore, we reviewed the genes and signaling pathways downstream of lncRNA-MALAT1, summarized its role in various cancers, and discussed the development of novel drugs targeting it.

LncRNAs can widely exist among mRNAs, partly owing to the similarity between their primary structures. For example, both contain exons and introns and have the same structures as 3′poly A and 5′caps. Furthermore, the substantial length of lncRNA facilitates the formation of relatively stable high-level structures (such as secondary structures). Therefore, it can participate in the organization and regulation of cells [5], enabling it to play various biological functions in the body. Thus, the mechanistic role of lncRNA in biological, physiological, and pathological processes may depend on its primary or secondary structure and participation in various processes of regulating cellular activities. Analyzing the structure of lncRNAs is an essential step in understanding their function and role.

Additionally, the lncALTAS database can be used to study the possible distribution of related lncRNAs and mRNAs in cancer cells. The relative concentration index (RCI) indicated the distribution of lncRNAs in the nuclear and cytoplasmic regions. The RCI is defined as the logarithmic ratio between the measured concentrations per kilogram base pair (FPKM) in two samples:$$RCI=\log_{2}(\frac{{Fraction}_{i}(RPKMs)}{{Fraction}_{j}(RPKMs)})$$
Note: RPKM = transcript reading per million maps, *i* = MALAT1 reading in the cytoplasm of the cell line, and *j* = MALAT1 reading in the cytoplasm of the cell line.

## 2. Function of LncRNA

In recent years, the functional mechanism of lncRNAs in the interaction between cancer-related fibroblasts and tumor cells has been summarized. LncRNAs can regulate signal transduction in cells, thereby promoting or inhibiting the proliferation of tumor cells, inducing their apoptosis, and ultimately regulating the growth and metastasis of tumor stem cells. LncRNAs can regulate and modify various gene expression processes by targeting the expression of multiple miRNAs and activating different signaling pathways associated with processes, such as epigenetics and gene regulation, before and after transcription.

### 2.1. Epigenetic Regulation

Epigenetics focuses on heritable gene expression changes without changing the nucleotide sequence of the gene. The known epigenetic phenomena include DNA methylation, genomic imprinting, maternal effects, gene silencing, nucleolar dominance, dormant transposon activation, and RNA editing [6].

LncRNAs were first used to study their epigenetic control based on genomic imprinting and the inactivation of the X chromosome. The structural basis of epigenetic control by lncRNAs was also investigated [7]. LncRNAs are key participants in cell differentiation, cell lineage selection, organogenesis, and tissue homeostasis. LncRNAs regulate gene expression to enable living cells to respond appropriately to changes in their external environment, thereby regulating metabolic processes and ultimately allowing the organisms to adapt better to these changes.

### 2.2. Transcriptional Regulation

LncRNAs can be regulated at the transcription level by interfering with the expression of adjacent genes or regulating the activity of transcription factors such as cofactors. For example, quantitative reverse transcription-polymerase chain reaction (qRT-PCR) was used to detect the expression of CTD-3252C9.4 in pancreatic cancer cells and tissues, and in vitro and in vivo functional experiments were used to identify the function of CTD-3252C9.4 in pancreatic cancer. FI6 was identified as a downstream target gene downregulated by CTD-3252C9.4, and its overexpression could antagonize the effect of CTD-3252C9.4 upregulation on the survival and apoptosis of pancreatic cancer cells [8]. These results suggest that lncRNAs could be used as transcription signals, and their transcription integrates developmental clues and reflects the appropriate intracellular environment. Therefore, the transcription of lncRNAs can be used as a signal to reflect the binding effect or signal pathway for transcription factors, and further clarify the spatiotemporal regulation of relevant lncRNAs, so that a regulatory direction is provided for subsequent steps of the metabolism.

Additionally, as a cofactor, lncRNA co-expresses with mRNA to treat the photodynamics of actinic keratosis. LncRNA can co-express with p53 and other immunohistochemistry proteins, thus interfering with gene transcription and translation [9]. This suggests that lncRNA can be used as a bait to bind to the target protein, thus preventing the target protein from approaching the chromatin and playing either a positive or negative role in regulating transcription. This mechanism might be universal for eukaryotic cells.

### 2.3. Post-Transcriptional Regulation

LncRNAs form double-stranded complexes with other RNAs after transcription to regulate gene expression. Owing to various restrictive factors, such as the free rotation of lncRNA molecules, the associated groups exhibit phenomena, such as diastereoisomerism, which are attributed to different spatial arrangements of the molecules; this enables the lncRNAs to play different roles. Therefore, lncRNAs can be used as scaffolds to combine domains with multiple effectors. 

It is possible to combine effectors with either transcriptional activation or inhibition activity. For example, lncRNA can be used as a molecular scaffold to modify histones, thereby contributing to tumor pathology. Furthermore, an lncRNA closely related to the tumor epidemic prevention and subsequent treatment (HOTAIR [10]) required at least two histone modification complexes to participate in histone modification. This stabilized the nucleic acid structure or signal complex, thus affecting transcriptional regulation.

Furthermore, AdipoQ AS lncRNA interacts with AdipoQ mRNA to form a double-stranded complex [11]. This complex restricts the free movement of AdipoQ mRNA, thus inhibiting the transfer of AdipoQ mRNA from the nucleus to the cytoplasm (as shown in Figure 1), indicating that lncRNAs can guide the transcription and expression of genes, such as mRNA transport. LncRNAs can also affect the transport process and ultimately regulate various physiological functions of cells by binding to different mRNAs.

## 3. The Regulatory Process of LncRNA-MALAT1 in Cancer Cells

Cancer is a complex disease involving multiple gene mutations, including epigenetic changes, chromosomal translocations, deletions, and increases. The genome encodes lncRNAs, but most are not translated into proteins. Although there is no translation, lncRNAs have crucial physiological functions in various cells, such as regulating chromatin, gene expression, growth, differentiation, and development [12]. LncRNAs regulate various pathophysiological processes in lung adenocarcinoma-related cells, such as proliferation, diffusion, metastasis, and apoptosis (as shown in Figure 2).

### 3.1. LncRNA-MALAT1 Affects Cell Proliferation and Apoptosis

RNA is transcribed and translated within the cell, which works in tandem with the cytoplasmic division for cell proliferation. Cell proliferation is necessary to maintain normal tissue development; however, mutated genes in cancer cells can cause uncontrolled malignant cell proliferation. A comparison of the expression profile of lncRNA MALAT1 (located on chromosome 10) in cancer and paraneoplastic tissues from 34 patients with non-small cell lung cancer indicated its expression in lung cancer A549 cells and its role in inhibiting the proliferation, invasion, and migration of the cells, eventually causing apoptosis in some cells [13].

lncRNA-MALAT1 also competes with endogenous miRNAs to regulate the expression levels of downstream genes. It controls the proliferation, invasion, metastasis, and apoptosis of lung cancer cells through various signaling pathways, such as PI3K/Akt and Wnt/β-catenin [14]. siRNA treatment in SW480 caused downregulation of lncRNA-MALAT1 gene expression, and the number of apoptotic cells and apoptosis rate increased 24 h after transfection. In contrast, the expression of Wnt and β-catenin in MALAT1 siRNA-transfected SW480 cells was inhibited [15]. Thus, lncRNA-MALAT1 can act on various signaling pathways and regulate their downstream genes to promote cancer cell growth and proliferation, leading to pathophysiological responses such as unlimited tumor proliferation in the organism.

### 3.2. LncRNA-MALAT1 Affects Cell Migration and Movement

Cell movement is one of the most critical outcomes in the evolution of life. Metastasis is a characteristic feature in malignant tumors, wherein cancer cells spread from their primary tumor site (via other routes) and continue to grow at a distant site, forming the same type of tumor as the primary site. lncRNA-MALAT1 could reduce the ability to regulate cancer cell migration by upregulating the expression of the target gene microRNA-432-5p (miR-432-5p) [16]. Furthermore, blocking the lncRNA-MALAT1 pathway could regulate EMT and counteract tumor metastasis [17].

Further investigation into the mechanistic role of lncRNA-MALAT1 revealed that the proportion of myeloid-derived suppressor cells (MDSCs), with high levels of immunosuppressive function and the related molecule arginase-1, was increased in the peripheral blood mononuclear cells of patients with lung cancer, whereas their lncRNA-MALAT1 levels were reduced. This suggests that the relative expression of lncRNA-MALAT1 is moderately and negatively correlated with the proportion of MDSCs. The results of in vitro experiments indicated that the proportion of MDSCs increased significantly after lncRNA-MALAT1 knockdown, and this is the first evidence that lncRNA-MALAT1 negatively regulates MDSCs [18]. Reducing lncRNA-MALAT1 levels results in an increased chance of cancer cell migration and disease progression, suggesting that lncRNA-MALAT1 as a lung cancer marker may be a potential strategy for treating metastatic lung cancer.

These results suggest that lncRNA-MALAT1 should be regarded as a cancer-related metastatic biomarker that can interfere with prevention treatments for cancer cell metastasis and provide a basis for further investigation of the mechanistic role of lncRNA-MALAT1 in various cancer phenotypes. The potential therapeutic role of lncRNA-MALAT1 in preventing metastasis can be further evaluated by setting up controlled experiments that knockdown the lncRNA-MALAT1 gene in animals while comparing the physiological characteristics of experimental and control animals and analyzing the pathophysiological indicators associated with the experimental animals.

### 3.3. LncRNA-MALAT1 Induces an Inflammatory Response in the Body

Inflammation is the defensive response of the body to external or internal stimuli and is a self-reactive function of the body. lncRNA-MALAT1 is elevated in the plasma of patients with acute myocardial infarction and may be involved in the inflammatory process as a regulator of other systems [19]. Downregulation of lncRNA-MALAT1 expression attenuates OGD/R-induced PC12 cell injury [20]. This mechanism may involve the attenuation of the inflammatory response and the inhibition of apoptosis. In addition, downregulation of lncRNA-MALAT1 expression inhibits the nuclear translocation of β-catenin, which leads to a decrease in c-Myc and MMP-7 levels [21], suggesting that it modulates inflammation and natural immunity. Moreover, lncRNA-MALAT1 can significantly attenuate myocardial injury in rats with polystyrene microsphere-induced pulmonary embolisms by inhibiting the MALAT1/NLRP3 signaling pathway [22]. lncRNA-MALAT1 is hypothesized to cause tissue damage and myocardial injury by regulating its downstream MALAT1/NLRP3 pathway, further confirming the vital role of lncRNA-MALAT1 in the LPS-induced inflammatory response.

These results suggest that lncRNA-MALAT1 plays a role in inflammation and immune regulation, which may exacerbate inflammation and tissue damage in organisms. Therefore, lncRNA-MALAT1 can be a novel drug target for treating various inflammatory diseases. The potential role of lncRNA-MALAT1 in treating inflammatory diseases was further evaluated by analyzing lung pathology and related pathophysiological parameters after administering MALAT1 inhibitors to experimental and control mice.

### 3.4. LncRNA-MALAT1 Competitively Binds to miRNA Sites

Some lncRNAs have sites that bind to miRNAs and can compete with miRNA target genes for regulating the expression of miRNAs and their target genes. These lncRNAs are miRNA sponges with adsorption properties, also known as competing endogenous RNAs (CompetingendousRNA, CeRNA). Various miRNAs were positively correlated with the expression of lncRNA-MALAT1. One study showed that lncRNA-MALAT1 acted as a ceRNA that adsorbed miRNA-384 on the NFKBIA gene to directly promote apoptosis of meningioma cells [23]. Thus, it inhibits the proliferation of cancer cells and promotes apoptosis to stop further cancer progression.

Correlation analysis identified SNHG6 and MALAT1 as upstream lncRNAs of hsa-miR-101-3p. Furthermore, the expression of Zeste homolog 2 (EZH2) was significantly associated with the infiltration of multiple immune cell types in hepatocellular carcinomas [24]. This suggests that lncRNAs can regulate EZH2 enhancement via spongy miRNAs. In addition, cyclic RNAs also regulate EZH2 signaling by targeting miRNAs similar to those of lncRNAs [25].

## 4. The Role of LncRNA-MALAT1 in Different Cancers

LncRNAs are closely associated with the generation and development of 12 types of cancer, including prostate, breast, lung, liver, pancreatic, kidney, colorectal, leukemia, gastric, ovarian, and bladder cancers, as well as cerebral edema. lncRNAs may exhibit tumor-suppressive and promotive (oncogenic) functions [26]. 

Figure 3 summarizes the physiological features of lncRNA-MALAT1 in the associated cancers and corroborates the dual role of MALAT1 in promoting unlimited cell proliferation and inhibiting cell growth at different sites. It also demonstrates the strong regulatory ability of lncRNA-MALAT1 to invade and metastasize in tissues. In recent years, MALAT1, the “star molecule” of lncRNAs, has been comprehensively studied, and its mechanism of action in various cancers has been clarified.

LncRNA-MALAT1 plays an essential role in cancers, such as nasopharyngeal, laryngeal, and gynecological cancers. As shown in Figure 4, lncRNA-MALAT1 can act on multiple functional pathways, suggesting that it can be expressed at various targets in different cell lines, thus exerting different mechanisms of action and producing other pathophysiological phenomena in the body.

### 4.1. Role of LncRNA-MALAT1 in Lung Cancer

Lung cancer is a severe respiratory disease characterized by coughing, shortness of breath, and, in severe cases, dyspnea and respiratory failure, which affect the quality of human life. The incidence and mortality of lung cancer are increasing yearly owing to an aging population, unhealthy lifestyles, and increased levels of atmospheric pollution [27]. According to relevant data, 9.96 million cancer deaths were recorded worldwide in 2020, with 2.2 million lung cancer cases ranking second and 1.8 million deaths ranking first, among malignant tumors [28]. Lung cancer is classified into two major groups: small-cell lung cancer and non-small-cell lung cancer, based on their degree of differentiation, morphological features, and biological characteristics [29]. The occurrence of lung cancer is closely related to the stimulation of adverse external environments, prolonged smoking, and recurrent inflammatory lesions in lung tissues. However, its pathogenesis is not yet fully understood. An in-depth study on the pathogenesis of lung cancer is of paramount significance to its diagnosis and treatment, as well as the development of drugs.

In a clinical study, researchers collected samples of tumors and the surrounding normal tissue from 34 patients with lung cancer who underwent surgery [30]. Through analysis of the samples, they found that lncRNA-MALAT1 was highly expressed in lung cancer tissue and significantly higher than in the surrounding normal tissue. In addition, other researchers used fluorescent quantitative PCR to detect plasma samples from 60 patients with NSCLC and 60 patients with benign lung disease [31] and observed high lncRNA-MALAT1 expression levels in the serum. Due to the increased expression levels of lncRNA-MALAT1 in both serum and cancer tissues, lncRNA-MALAT1 in the blood of tumor patients may be assumed to be released by cancer tissues via exosomes into the blood. Therefore, its expression in the blood parallels that of the primary tumors [32]. This suggests that lncRNA-MALAT1 could be used as a biomarker for detecting lung cancer in its early stages. It could also be an effective tool for monitoring cancer progression and treatment efficacy.

Numerous experimental studies have revealed that the mechanism of lncRNA-MALAT1 involvement in lung cancer development is closely related to cellular activities that affect the proliferation, motility, and development of cancer cells. MALTA1-miRNA regulatory pathways associated with lung cancer are summarized in Figure 5 [33,34,35,36,37,38,39]. Most studies on the role of lncRNA-MALTA1 in lung cancer have focused on further influencing the level of specific miRNA target genes through the competitive binding of specific miRNAs, which provides a research idea and basis for revealing the pathogenesis of lung cancer.

### 4.2. Role of LncRNA-MALAT1 in Nasopharyngeal and Laryngeal Cancers

Nasopharyngeal carcinoma is a generic term for malignant epithelial tumors with different etiologies and extensive histopathological manifestations [40]. To investigate the expression and biological functions of lncRNA-MALAT1 in nasopharyngeal carcinoma cell lines, lncRNA-MALAT1 lentiviral interference and activation vectors were constructed and stably transfected with the nasopharyngeal carcinoma cell line CNE-1 using RNAi and RNA activation technologies. The effect of lncRNA-MALAT1 on the biological behavior of CNE-1 cells was analyzed, and the upregulation of the lncRNA-MALAT1 gene promoted the proliferation, invasion, and metastatic ability of CNE-1 cells in nasopharyngeal carcinoma [41]. The main causative factors in the development of nasopharyngeal carcinoma are genetic susceptibility, EBV infection, and other environmental factors [42], and few studies have explored the role of lncRNA-MALAT1. Hence, little is known about its mechanism of action and relative therapeutic approaches.

Laryngeal cancer is a malignant tumor that occurs in the larynx and accounts for approximately 1–5% of the incidence of malignant tumors throughout the body. The incidence and mortality rate of laryngeal cancer in China is low to medium worldwide, with significant urban-rural and gender differences [43]. The inhibition of high-level expression of lncRNA-MALAT1 in laryngeal squamous carcinoma transplantation tumor cells (using the lncRNA-MALAT1 siRNA lentiviral expression vector) significantly induced apoptosis and autophagy [44]. This suggests that lncRNA-MALAT1 may play a role in laryngeal cancer. However, this finding requires further exploration and is expected to be a target for gene therapy in laryngeal cancer and other malignancies.

The expression levels of lncRNA-MALAT1, miRNA-503-5p, and FOXK1 were examined in the LC cells. The levels for lncRNA-MALAT1 and FOXK1 were high, but the level of miR-503-5p was low. This suggests that miR-503-5p is one of the target regulatory genes of lncRNA-MALAT1 and that the two are negatively correlated. The experimental downregulation of miR-503-5p significantly promoted the proliferation and invasion abilities of the LC cells. It also reversed the inhibitory effect of the low levels of expression of lncRNA-MALAT1 on the proliferation and invasion of LC cells [45]. Thus, the down-regulation of lncRNA-MALAT1 expression could alleviate disease symptoms and ultimately have the therapeutic effect of inhibiting lung fibrosis.

### 4.3. The Role of LncRNA-MALAT1 in Gynecological Cancers

Gynecological cancer is a significant threat to modern life and poses a serious risk to the health of women. These cancers mainly include breast, ovarian, uterine, vaginal, cervical, and vulvar [46]. LncRNA-MALAT1 is a potential biomarker for disease diagnosis and prognosis and a candidate for drug targeting in treating gynecological tumors [47].

In breast cancer experiments, researchers used short hairpin RNA (shRNA), a type of lncRNA-MALAT1, to knock down endogenous lncRNA-MALAT1 in human breast cancer (MCF-7) cells. They showed that this significantly inhibited cell proliferation, migration, and tubular cell formation, which would have promoted further development of breast cancer.

In contrast to the low levels of MiR-145 expression in BC tissues, downregulation of endogenous lncRNA-MALAT1 increased it significantly in MCF-7 cells. This finding suggests an interaction between lncRNA-MALAT1 and miR-145. In addition, the knockdown of lncRNA-MALAT1 significantly decreased the expression of vascular endothelial growth factor in MCF-7 cells. lncRNA-MALAT1 promotes breast cancer angiogenesis, which may be related to miR-145 expression levels [48]. As shown in Figure 6, data collected from the lncATLAS database were analyzed, with the help of RCI, for lncRNA-MALAT1 in MCF-7 human breast cancer cell lines, in terms of CN RCI values as well as shaded areas. Results showed that the intensity of localization in the nucleus was higher than that in the cytoplasm, indicating that lncRNA-MALAT1 was more actively expressed in the nucleus of human breast cancer cells. LncRNA-MALAT1 was strongly localized in the nucleus compared with other parts of the cell, and the nucleus controls cell growth and development. Therefore, lncRNA-MALAT1 positively correlates with the proliferation and growth of MCF-7 cells, promoting the growth and development of human breast cancer. These findings provide a theoretical basis for the mechanistic study of the association between lncRNA-MALAT1 and specific cancers, which can later be used in experiments with mice and MALAT1 inhibitors. One should be able to compare the experimental phenomena with control groups to provide effective and reliable clinical treatment options for human breast cancer.

Additional experiments revealed that lncRNA-MALAT1 expression was upregulated in cervical cancer cell lines compared with that of normal cervical squamous cell samples. Further studies on the effect of lncRNA-MALAT1 on cell phenotypes revealed that lncRNA-MALAT1 plays a role in promoting cell migration and proliferation. Notably, in CaSki cells, lncRNA-MALAT1 expression decreased with HPV16E6/E7 gene knockdown [49]. Furthermore, studies on clinical specimens indicated that lncRNA-MALAT1 is expressed in HPV-positive cervical squamous cells but not in HPV-negative, normal cervical squamous cells. These results suggest that in cervical cancer, HPV is associated with losing control of the lncRNA-MALAT1 gene.

Drugs targeting DNA repair defects and angiogenesis treat advanced or recurrent ovarian and cervical cancers. Immune checkpoint inhibitors under development, such as anti-PD-1/PD-L1 antibodies, are effective in treating mismatch repair-deficient endometrial cancers and HPV-associated malignancies [50]. As medical technology evolves, scientists use their understanding of cancer biology and genomics to develop additional predictive biomarkers for targeted therapies and to maximize patient benefits.

### 4.4. Role of MALAT1 in Other Diseases

LncRNAs regulate the proliferation and drug resistance of osteosarcoma. MALAT1 regulates the expression of the cell cycle protein-dependent kinase 9 (CDK9) by sponging miR-206, thereby controlling the progression of osteosarcoma [51]. The knockdown of the lncRNA-MALAT1 gene in the experiment resulted in the inhibition of the proliferation of osteosarcoma cells, suggesting that lncRNA-MALAT1 plays an oncogenic role in developing osteosarcoma. Although the role of lncRNA-MALAT1 in miR-206/CDK9 axis-mediated osteosarcoma proliferation remains unclear, the findings indicate that the MALAT1/miR-206/CDK9 axis may provide novel insights into the biological mechanisms of osteosarcoma progression.

The intratumoral injection of lncRNA-MALAT1 siRNA, which silences lncRNA-MALAT1, could retard the growth and reduce the metastasis of prostate cancer transplant tumors in desmoplastic, nude male mice, thereby prolonging the survival time of tumor-bearing mice [52]. This suggests that lncRNA-MALAT1 may be a potential therapeutic target for prostate and other cancers.

In addition, lncRNA-MALAT1 plays a vital role in body fluid tumors such as leukemia. For example, a controlled assay was performed to quantify the expression levels of lncRNA-MALAT1 in the peripheral blood of patients with acute myeloid leukemia (AML) sepsis. The results revealed that lncRNA-MALAT1 expression was upregulated in AML patients with sepsis. The overall survival rate was significantly lower in the group with high lncRNA-MALAT1 expression levels than in the group with low expression levels [53]. Artesunate, a derivative of artemisinin, could regulate the expression of apoptosis-related proteins such as Bcl-2, Bax, caspase-3, and PTEN via the PI3K/AKT signaling pathway and promote apoptosis in human AML cells. In contrast, lncRNA-MALAT1 could act on the PI3K/AKT pathway, thus affecting the expression of related proteins, as summarized earlier. This suggests that the upregulation of lncRNA-MALAT1 in AML patients with sepsis could negatively affect their clinical characteristics and survival by acting on the PI3K/AKT pathway [54]. LncRNA-MALAT1 may also be an independent prognostic factor in AML sepsis and may become a putative diagnostic marker and therapeutic target for patients with AML sepsis.

There is growing evidence that hypoxia also contributes to many human diseases [55], such as the post-stroke blood-brain barrier (BBB), ischemic heart muscle, and heart failure (HF). The upregulation of lncRNA-MALAT1 reduces cytotoxicity and apoptosis after oxygen and glucose deprivation (OGD), whereas the opposite occurs after lncRNA-MALAT1 silencing. Some experiments have used the activation of the MALAT1/CREB/PGC-1α/PPARɣ signaling pathway to assess its protective effect (from ischemic injury) on vascular endothelial cells [56]. Ischemic heart disease, a major cause of heart failure, is closely related to hypoxia. In a CoCl2-induced hypoxia model of CSCs, the MALAT1/miR-155/MEF2A pathway was affected by miR-155 sponge adsorption, which consequently regulates the expression of MEF2A and ultimately promotes cell proliferation and migration [57].

## 5. New Drug Development Using lncRNA-MALAT1 as a Target

LncRNA-MALAT1 was one of the first lncRNAs to be studied in cancer metastasis and is predictive of the survival of patients with early-stage cancers. Preclinical models of metastatic tumors have shown potential as therapeutic targets for cancer treatment. Suppose antisense oligonucleotides (ASOs) are used in vivo to bind to target mRNAs via base complementary pairing, resulting in RNAseH-mediated knockdown of lncRNA-MALAT1 as a target for silencing. In this case, it can also inhibit gene expression by, for example, altering mRNA shearing and inhibiting ribosomal translation to attenuate metastasis in mouse models [58].

In addition, relevant experiments accurately identified the MALAT1 inhibitor, a therapeutic agent targeting lncRNA-MALAT, and for the first time, the rationale for the application of this inhibitor in the prevention and treatment of abdominal aortic aneurysms was discovered [59]. LncRNA-MALAT1 inhibitor, MALAT1-IN-1, can protect the aorta by preventing, inhibiting, and reversing abdominal aortic aneurysms based on three aspects. Thus, the novel application of the lncRNA-MALAT1 inhibitor has significant clinical value in abdominal aortic aneurysm treatment strategies.

Researchers also conducted a series of liver regeneration experiments in a mouse model with 2/3 of the liver removed, as well as in vitro functional analysis of lncRNA-MALAT1. It was found that lncRNA-MALAT1 activated the Wnt/β-catenin signaling pathway and promoted cyclin D1 expression by inhibiting the expression of AXIN1 and APC genes. Overall, the results of this study suggest that lncRNA-MALAT1 is a key molecule in liver regeneration, and that pharmacological interventions targeting lncRNA-MALAT1 may promote liver regeneration, thus facilitating the treatment of liver failure and transplantation [60].

LncRNA-MALAT1 reportedly plays a crucial role in cancer development and progression. The development of lncRNA-MALAT1-based targeting drugs and insights into their mechanisms of action are vital for cancer treatment.

## 6. Summary and Outlook

Gene variants of lncRNA-MALAT1 are associated with various cancers [61]. The regulation of cell-cycle-related transcription factor expressions promotes cell proliferation. This reduces the expression levels of related RNAs, which in turn leads to abnormal changes in the cell cycle, resulting in the cancerous transformation of normal tissue cells that would not normally proliferate and differentiate, ultimately causing further deterioration of the condition of patients with cancer. For patients with lung cancer, the expression characteristics of lncRNA-MALAT1 in serum can be used as a marker for diagnosis [62], whereas the expression level of lncRNA-MALAT1 has some value in the identification of the pathological types of lung cancer. In addition, the combination of lncRNAs in plasma with classical tumor markers, CEA or Cyfra21-1, can enhance the diagnostic efficacy of lncRNAs in patients with lung cancer [63]. This can be further studied or combined with other markers to improve the accuracy of a lung cancer diagnosis.

However, genes in cancer cells act through multiple signaling pathways. Thus, signaling in cancer cells is a multi-targeted and multi-linked regulatory process [64]. Therefore, it can be divided into two types of treatments: single- and multi-target therapies. Single-target inhibitors can only block one signaling pathway. Cancer cells can remedy or escape through other routes or even activate the rapid amplification of other tumor genes, ultimately leading to cancer recurrence, metastasis, and treatment failure. In contrast, the effect of multi-target inhibitors is superior to that of EGFR single-target inhibitors for cancer treatment [65], suggesting that multi-target inhibitors be further explored for lung cancer treatments.

Cancer is characterized by substantially high morbidity and mortality. Therefore, further experiments must be performed to elucidate the action mechanism of lncRNA-MALAT1, weigh the advantages and potential adverse effects of the signaling pathways associated with it, and select the optimal blocking method for cancer cell proliferation and migration. Furthermore, in conjunction with the translational mechanism of lncRNA-MALAT1 and its expression in lung cancer cells, it should be possible to reveal the relevance of lncRNA-MALAT1 polymorphisms in carcinogenesis. However, this hypothesis must be validated by specific experiments at a later stage, for example, by investigating the suppression of lncRNA-MALAT1 antimetabolites and evaluating the results by measuring the growth or spread of cancer cells.

Although this article summarizes the sites and pathways of action of lncRNA-MALAT, it does not provide a comprehensive summary of the therapeutic approaches to its expression in various body parts, which needs further exploration. Furthermore, this study suggests that targeted therapeutics could further be investigated to treat diseases caused by lncRNA-MALAT1 more accurately and effectively.

## Figures and Tables

**Figure 1 molecules-28-02126-f001:**
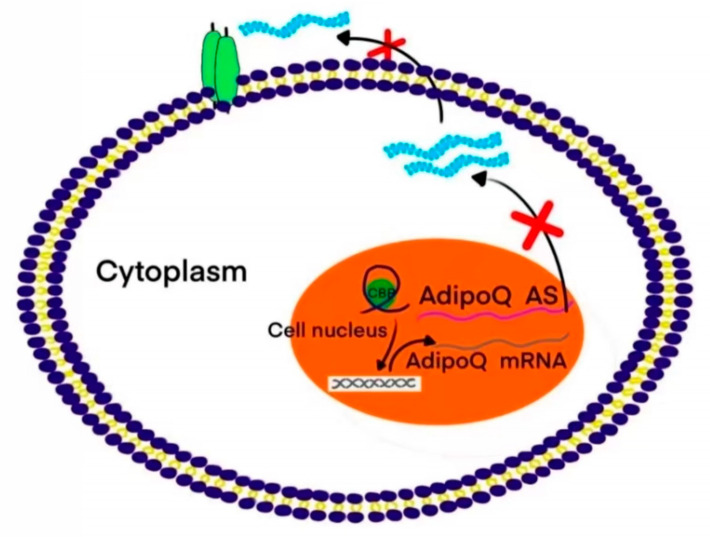
Interaction between lncRNA and mRNA double-stranded complexes.

**Figure 2 molecules-28-02126-f002:**
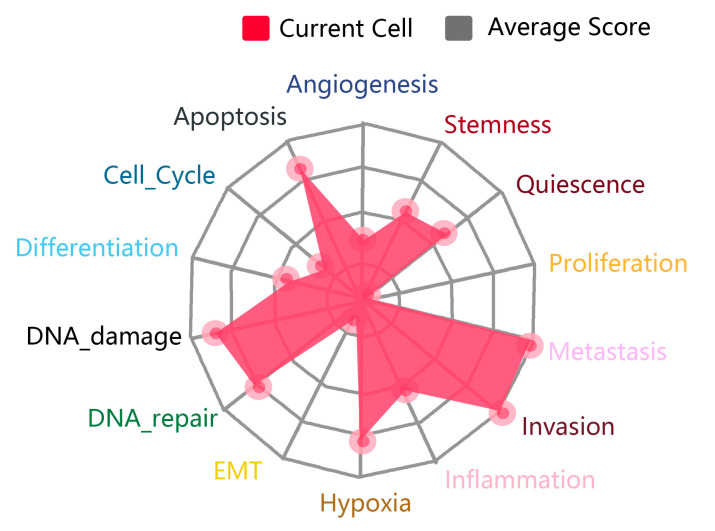
Pathophysiological processes associated with the presence of lncRNA in lung adenocarcinoma.

**Figure 3 molecules-28-02126-f003:**
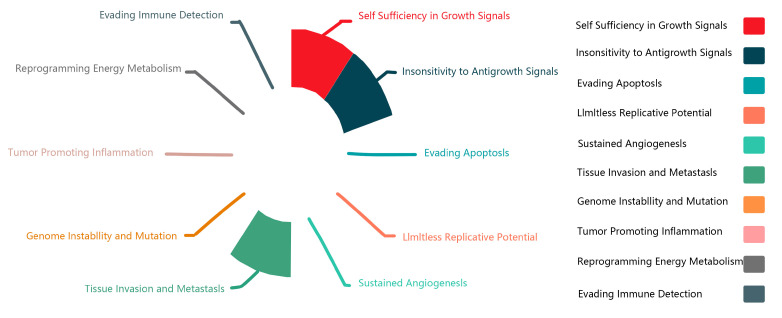
Physiological characteristics of MALAT1 in various types of cancer.

**Figure 4 molecules-28-02126-f004:**
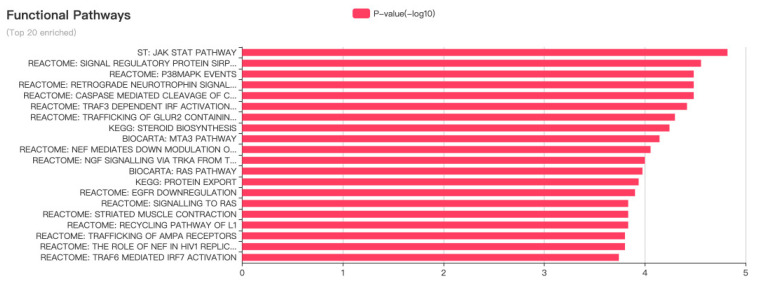
Ten important pathways through which MALAT1 functions.

**Figure 5 molecules-28-02126-f005:**
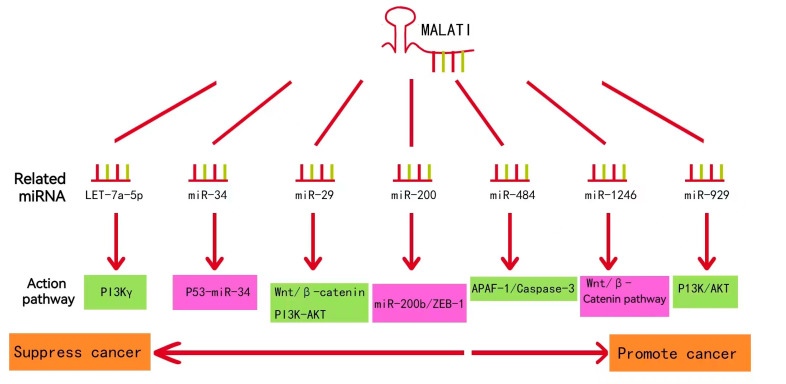
MALAT1-miRNA regulatory pathway associated with lung cancer.

**Figure 6 molecules-28-02126-f006:**
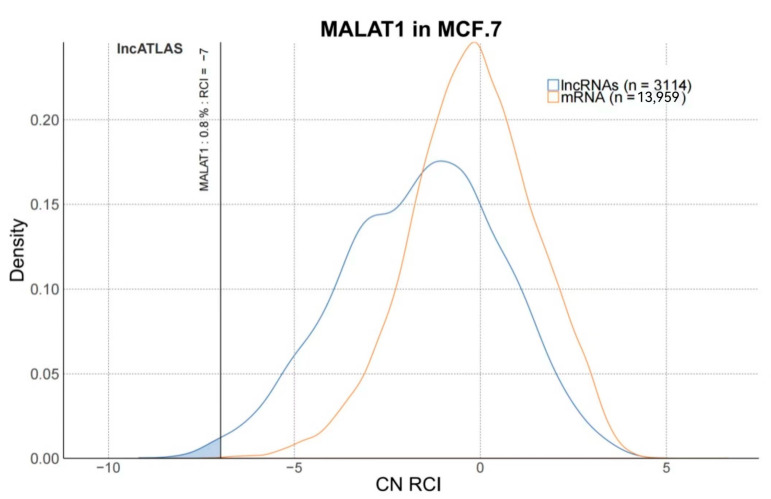
CN RCI values of MALAT1 in MCF.7.

## Data Availability

Not applicable.

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
