# Peer review of "LncRNA-MALAT1: A Key Participant in the Occurrence and Development of Cancer"

_molecules, 2023, doi:10.3390/molecules28052126_

Round 1
Reviewer 1 Report
In this review authors summarized the possible pathophysiological phenomena of lncRNA-MALAT1 in relevant cancers and its potential therapeuticapplications in these conditions for the discovery of new drugs. My major concern is the novelity since lncRNA-MALAT1 in cancers has been reviewed several times, besides the low quality of presenting and discussing information. In valuable results in this review should be repressented and rewritten before a future submission.
* Abstract: Why ithis review is very long and in an article like style (Background, methods, results)?
* Abstract: Background is long, long sentences
* Abstract: Methods , please split this one sentence.
* Abstract: Conclusion, (1st sentence: Long non coding RNA (lncRNA) is a group of non coding RNA transcripts longer than 200 nucleotides).
So here author are concluding that LncRNA is ....., I do not think that is away to present a review should be publishe at Molecules, I think it is not acceptable here!
** Subtitles: I do not know why the authors c used this style of short subtitiles.
i.e. 3.1 The role of lncRNA-MALAT1 in lung cancer
3.1.1Affect cell proliferation and apoptosis
3.1.2Influence on cell migration and movement
3.1.3Induced inflammatory response of the organism
3.1.4Competitive binding of miRNA
- In approprite style, no synchronization, Lccoding is a singularm so present verb should be ended with s,, what do authors mean with "Competitive binding of miRNA" not clear for me. even when I read in line 273: "LncRNA includes multiple miRNA binding sites, which are miRNA sponge adsorbers, similar to sponges that adsorb water, and can competitively bind miRNA to compete for endogenous RNA." It is not clear and scientificaly authors do not have to explain the sponges action and competiitively compete are not approprite expression.
3.4 Role of MALAT1 in other diseases " In recent years, it has been proved that lncRNA can regulate the proliferationanddrug resistance of osteosarcoma. It has been observed that MALAT1 regulates the expressionof cyclin dependent kinase 9 (CDK9) through sponge miR-206, thereby regulating the progression of osteosarcoma................" So this section is discussing only a single study, only one reference was cited, ?
In summary and outlock:
Ref 59: it is only about CC cancer and not various types?
Also there is only one reference in summary and outlock?
Author Response
Dear Editor:
Thank you very much for your letter and advice on our manuscript. We have carefully revised the manuscript according to reviewer’s suggestion. The revise are using a word processing program in the manuscript. We also used a paid editing service (Editage, www.editage.cn) to check the manuscript. We hope that the revision is acceptable and look forward to hearing from you soon.
Our responses to reviewer’s comments are listed below:
Point 1: Abstract: Why ithis review is very long and in an article like style (Background, methods, results)?
Response 1:
I'm sorry that I didn't understand the specific requirements before, which caused this part to be too long and confusing. We have revised this part in the abstract and revised as following:
“Abstract: LncRNAs are a group of non-coding RNA transcripts with lengths of over 200 nucleotides and can interact with DNA, RNA, and proteins to regulate gene expression of malignant tumors in human tissues. LncRNAs participate in vital processes, such as chromosomal nuclear transport in the cancerous site of human tissue, activation, as well as regulation of proto-oncogenes, differentiation of immune cells, and regulation of the cellular immune system. The lncRNA metastasis-associated lung cancer transcript 1 (MALAT1) is reportedly involved in the occurrence and development of many cancers and serves as a biomarker and therapeutic target. These findings highlight its promising role in cancer treatment. In this article, we comprehensively summarized the structure and functions of lncRNA, notably the discoveries of lncRNA-MALAT1 in different cancers, the action mechanisms, and the ongoing research on new drug development. We believe our review would serve as a basis for further research on the pathological mechanism of lncRNA-MALAT1 in cancer and provide evidence and novel insights into its application in clinical diagnoses and treatments.”
Point 2: Subtitles: I do not know why the authors c used this style of short subtitiles.
Response 2:
The subtitiles in the article have been more appropriately revised to give a more precise summary of meaning. Details of the changes can be found in the final draft.
We have revised as following:
“3.1 LncRNA-MALAT1 affects cell proliferation and apoptosis
.3.2 LncRNA-MALAT1 affects cell migration and movement
.3.3 LncRNA-MALAT1 induces inflammatory response of the body
.3.4 LncRNA-MALAT1 competitively binds to miRNA sites ”
Point 3: What do authors mean with "Competitive binding of miRNA" not clear for me. even when I read in line 273: "LncRNA includes multiple miRNA binding sites, which are miRNA sponge adsorbers, similar to sponges that adsorb water, and can competitively bind miRNA to compete for endogenous RNA." It is not clear and scientificaly authors do not have to explain the sponges action and competiitively compete are not approprite expression.
Response 3:
Regarding the "sponge adsorption" in 3.4, I've revised it in a relatively clearer way.
We have revised this part in the manuscript as following: “Some lncRNAs have sites that bind to miRNAs and can compete with miRNA target genes for regulating the expression of miRNAs and their target genes. These lncRNAs are miRNA sponges, also known as competing endogenous RNAs (CompetingendousRNA, CeRNA), with adsorption effects. Various miRNAs were positively correlated with the expression of lncRNA-MALAT1. One study showed that lncRNA-MALAT1 acted as a ceRNA that adsorbed miRNA-384 on the NFKBIA gene to directly promote apoptosis of meningioma cells [23]. Thus, it inhibits the proliferation of cancer cells and promotes apoptosis to stop further cancer progression.”
Point 4: 3.4 Role of MALAT1 in other diseases " In recent years, it has been proved that lncRNA can regulate the proliferationanddrug resistance of osteosarcoma. It has been observed that MALAT1 regulates the expressionof cyclin dependent kinase 9 (CDK9) through sponge miR-206, thereby regulating the progression of osteosarcoma................" So this section is discussing only a single study, only one reference was cited? Also there is only one reference in summary and outlook? In summary and outlock: Ref 59: it is only about CC cancer and not various types? Also there is only one reference in summary and outlock?
Response 4:
Thank you for clearly pointing out that reference 59 and the summary and outlook section are too poorly cited, which makes me realise that the presentation is indeed unclear and unconvincing. So I have added further literature to round out the entire article and make this article more complete and specific.The details are likewise reflected in the article.
We have revised this part in the manuscript as following: “LncRNAs regulate the proliferation and drug resistance of osteosarcoma. MALAT1 regulates the expression of the cell cycle protein-dependent kinase 9 (CDK9) by sponging miR-206, thereby controlling the progression of osteosarcoma [51]. The knockdown of the lncRNA-MALAT1 gene in the experiment resulted in the inhibition of the proliferation of osteosarcoma cells, suggesting that lncRNA-MALAT1 plays an oncogenic role in developing osteosarcoma.”
And “Gene variants of lncRNA-MALAT1 are associated with various cancers [61]. The regulation of cell-cycle-related transcription factor expressions promotes cell proliferation. This reduces the expression levels of related RNAs, which in turn leads to abnormal changes in the cell cycle, resulting in the cancerous transformation of normal tissue cells that would not normally proliferate and differentiate, ultimately causing further deterioration of the condition of patients with cancer. For patients with lung cancer, the expression characteristics of lncRNA-MALAT1 in serum can be used as a marker for diagnosis [62], whereas the expression level of lncRNA-MALAT1 has some value in the identification of the pathological types of lung cancer. In addition, the combination of lncRNAs in plasma with classical tumor markers, CEA or Cyfra21-1, can enhance the diagnostic efficacy of lncRNAs in patients with lung cancer [63]. This can be further studied or combined with other markers to improve the accuracy of a lung cancer diagnosis.
However, genes in cancer cells act through multiple signaling pathways. Thus, signaling in cancer cells is a multi-targeted and multi-linked regulatory process [64]. Therefore, it can be divided into two types of treatment: single- and multi-target therapies. Single-target inhibitors can only block one signaling pathway. Cancer cells can remedy or escape through other routes or even activate the rapid amplification of other tumor genes, ultimately leading to cancer recurrence, metastasis, and treatment failure. In contrast, the effect of multi-target inhibitors is superior to that of EGFR single-target inhibitors for cancer treatment [65], suggesting that multi-target inhibitors be further explored for lung cancer treatments.”
We have slightly restructured the article as a whole and added examples of leukaemia to the section "4.4 Role of MALAT1 in other diseases" to discuss the role of lncRNA-MALAT1 in liquid tumours, adding variety to the specific examples. In addition, we have further polished and added to the article to address the poor quality of the presentation of information and discussion in the article.
We have revised this part as following: “In addition, lncRNA-MALAT1 plays a vital role in body fluid tumors such as leukemia. For example, a controlled assay was performed to quantify the expression levels of lncRNA-MALAT1 in the peripheral blood of patients with acute myeloid leukemia (AML) sepsis. The results revealed that lncRNA-MALAT1 expression was upregulated in AML patients with sepsis. The overall survival rate was significantly lower in the group with high lncRNA-MALAT1 expression levels than in the group with low expression levels [53]. Artesunate, a derivative of artemisinin, could regulate the expression of apoptosis-related proteins such as Bcl-2, Bax, caspase-3, and PTEN via the PI3K/AKT signaling pathway and promote apoptosis in human AML cells. In contrast, lncRNA-MALAT1 could act on the PI3K/AKT pathway, thus affecting the expression of related proteins, as summarized earlier. This suggests that the upregulation of lncRNA-MALAT1 in AML patients with sepsis could negatively affect their clinical characteristics and survival by acting on the PI3K/AKT pathway [54]. LncRNA-MALAT1 may also be an independent prognostic factor in AML sepsis and may become a putative diagnostic marker and therapeutic target for patients with AML sepsis.”
Once again, thank you very much for comments and suggestions!
Sinserely!
Reviewer 2 Report
The manuscript titled “LncRNA -MALAT1:a key participant in the occurrence and development of cancer” by Hao et al., is an interesting manuscript. The authors start by providing information on LncRNA Structure, Function of lncRNA, Epigenetic regulation, and its Transcriptional and Post-transcriptional regulation. Thea authors further discuss the role of lncRNA-MALAT1, The role of lncRNA-MALAT1 in lung cancer, Effects of lncRNA-MALAT1 on cell proliferation and apoptosis, Influence of lncRNA-MALAT1 on cell migration and movement, lncRNA-MALAT1 effects on inflammatory response. The authors then discuss the implications of lncRNA-MALAT1 in nasopharyngeal and laryngeal carcinoma, gynecological cancers, as well as other diseases such as prostate cancer and osteosarcoma. Lastly, the authors discuss New drug development using lncRNA-MALAT1 as a target. The authors have used adequate number of relevant references to support their claims. The manuscript is well-organized and contains adequate amount of information.
Author Response
Dear Editor:
Thank you very much for your letter and advice on our manuscript. We have carefully revised the manuscript according to reviewer’s suggestion. The revise are using a word processing program in the manuscript. We also used a paid editing service (Editage, www.editage.cn) to check the manuscript. We hope that the revision is acceptable and look forward to hearing from you soon.
Our responses to reviewer’s comments are listed below:
Point: The manuscript titled “LncRNA -MALAT1:a key participant in the occurrence and development of cancer” by Hao et al., is an interesting manuscript. The authors start by providing information on LncRNA Structure, Function of lncRNA, Epigenetic regulation, and its Transcriptional and Post-transcriptional regulation. Thea authors further discuss the role of lncRNA-MALAT1, The role of lncRNA-MALAT1 in lung cancer, Effects of lncRNA-MALAT1 on cell proliferation and apoptosis, Influence of lncRNA-MALAT1 on cell migration and movement, lncRNA-MALAT1 effects on inflammatory response. The authors then discuss the implications of lncRNA-MALAT1 in nasopharyngeal and laryngeal carcinoma, gynecological cancers, as well as other diseases such as prostate cancer and osteosarcoma. Lastly, the authors discuss New drug development using lncRNA-MALAT1 as a target. The authors have used adequate number of relevant references to support their claims. The manuscript is well-organized and contains adequate amount of information.
Response:
I would like to thank you very much for your willingness to take the valuable time to professionally review my article. I feel very honoured. And by reviewing the content and structure of my article again, further refinements and additions have been made to this article, hopefully making it even better. For example:
1.We changed the abstract to be more in line with the requirements of the manuscript.
2.The subheadings of the article were more accurately condensed.
3.We have also added more accurate explanations of the technical terms in the article.
4.In this section of 4.4 we have added an analysis on liquid tumour-leukaemia to further refine the mechanism of action of lncRNA-MALAT1 in various types of cancer
- At the end of the article we have added relevant literature to support the conclusions of this article to make it more convincing.
Thank you once again for your acknowledgement of the content of this article. We will continue to work hard!
Sincerely!
Reviewer 3 Report
The authors have analyzed the role of lnc RNA MALAT1 in several types of cancer. However, some minor things could be addressed in the review to be in an acceptable format.
The authors could discuss MALAT1 lnc RNA in liquid tumors- leukemia and lymphoma if there is literature available.
The authors could discuss the role of MALAT1 with circular RNAs in cancer.
The overall figure resolution could be improved.
Author Response
Dear Editor:
Thank you very much for your letter and advice on our manuscript. We have carefully revised the manuscript according to reviewer’s suggestion. The revise are using a word processing program in the manuscript. We also used a paid editing service (Editage, www.editage.cn) to check the manuscript. We hope that the revision is acceptable and look forward to hearing from you soon.
Our responses to reviewer’s comments are listed below:
Point 1: The authors could discuss MALAT1 lnc RNA in liquid tumors- leukemia if there is literature available. And the authors could discuss the role of MALAT1 with circular RNAs in cancer.
Response 1:
Thank you very much for your suggestion. After further review of data, we have added an example of leukemia in the section "4.4 Role of MALAT1 in other diseases" to discuss the role of lncRNA-MALAT1 in liquid tumors, which adds diversity to specific examples. In addition, we have further polished and supplemented the article to solve the problem of poor quality of information and discussion in the article.
We have revised this part as following: “In addition, lncRNA-MALAT1 plays a vital role in body fluid tumors such as leukemia. For example, a controlled assay was performed to quantify the expression levels of lncRNA-MALAT1 in the peripheral blood of patients with acute myeloid leukemia (AML) sepsis. The results revealed that lncRNA-MALAT1 expression was upregulated in AML patients with sepsis. The overall survival rate was significantly lower in the group with high lncRNA-MALAT1 expression levels than in the group with low expression levels [53]. Artesunate, a derivative of artemisinin, could regulate the expression of apoptosis-related proteins such as Bcl-2, Bax, caspase-3, and PTEN via the PI3K/AKT signaling pathway and promote apoptosis in human AML cells. In contrast, lncRNA-MALAT1 could act on the PI3K/AKT pathway, thus affecting the expression of related proteins, as summarized earlier. This suggests that the upregulation of lncRNA-MALAT1 in AML patients with sepsis could negatively affect their clinical characteristics and survival by acting on the PI3K/AKT pathway [54]. LncRNA-MALAT1 may also be an independent prognostic factor in AML sepsis and may become a putative diagnostic marker and therapeutic target for patients with AML sepsis.”
Point 2:The overall figure resolution could be improved.
Response 2:
Regarding the graphics issue you pointed out, we have also consulted with others to improve the overall graphics resolution.
Thank you again for taking the time to review and advise on this article, we'll keep working on it!
Sinserely!
Round 2
Reviewer 1 Report
My comments were sent to the editor
Author Response
Thank you very much for your comments and suggestions, which have made our articles better and more logical, and we will continue to work hard to write more innovative and valuable articles for our readers to read. Thank you again!